# Electrochemical Behavior of Rejected Dental Implants in Peri-Implantitis

**Ioana Bunoiu [1,†], Mihai Andrei [1,†], Cristian Scheau [2,*], Claudiu Constantin Manole [3],**
**Andrei Bogdan Stoian [3], Vladimir Sorin Ibric Cioranu [4] and Andreea Cristiana Didilescu [1,*]**

1    Division of Embryology, Faculty of Dental Medicine, Carol Davila University of Medicine and Pharmacy,
     050474 Bucharest, Romania; bunoiu.ioana@gmail.com (I.B.); dr.andrei.mihai@gmail.com (M.A.)
2    Department of Physiology, Faculty of Medicine, Carol Davila University of Medicine and Pharmacy,
     050474 Bucharest, Romania
3    Department of General Chemistry, Faculty of Applied Chemistry and Materials Science, University
     Politehnica of Bucharest, 011061 Bucharest, Romania; claudiu.manole@upb.ro (C.C.M.);
     andreibstoian@yahoo.com (A.B.S.)
4    Division of Oral Maxillofacial Surgery, Faculty of Dental Medicine, Titu Maiorescu University,
     040441 Bucharest, Romania; isorin83@yahoo.com
*    Correspondence: cristian.scheau@umfcd.ro (C.S.); andreea.didilescu@umfcd.ro (A.C.D.);
     Tel.: +40-722536798 (A.C.D.)
†    The authors contributed equally to this work.

**Abstract:** This paper aims to investigate the electrochemical stability of four dental implants based on titanium alloys, rejected and recovered from patients diagnosed with peri-implantitis. The recovered implants were investigated over one week through open circuit potential (OCP) measurements, Tafel analysis, electrochemical impedance spectroscopy (EIS), and scanning electron microscopy (SEM) coupled with energy dispersive spectrometry (EDS). Patients' X-rays and clinical data were collected. The OCP measurements showed numerous variations of the potential over time, with increases and decreases, which indicated passivation–depassivation cycles. The main corrosion parameters were obtained through Tafel analysis. Corrosion rates and polarization curves suggested a greater instability trend over time for one implant. Bode Modulus and Nyquist diagrams were obtained after EIS was performed and electrical circuits were proposed and fitted for the dental implants in order to follow the materials resistance over time. Although two implants had the highest initial recorded resistances, they showed the most decrease over time. SEM micrographs showed pitting corrosion, while the presence of the Cl element in the EDS spectra indicated the presence of chlorides associated with these processes. The analyses performed on the dental implants denoted instability, with a different behavior for each one.

**Keywords:** corrosion; EDS; EIS; OCP; SEM; titanium alloy

---

## 1. Introduction

Since their introduction by Brånemark in the 1960s, oral implants have become an increasingly used option in dental practices for restoring dento-maxillary functions by replacing teeth that were lost through trauma or various pathological processes [1,2]. The clinical results [3] and prognosis [4] of implant therapy are influenced by the patient's alveolar bone, regulated by local and systemic factors, but also by the physical and chemical properties of the implanted materials [5]. These properties include the implant microstructure and the composition and characteristics of its surfaces [6]. Thus, an ideal implant material should be biocompatible and resistant to corrosion, wear, and fractures [7,8]. With such requirements, Ti alloys are the most-used dental materials for implant works [9].

To obtain a good stability of dental reconstructions [10], all the factors that contribute to the oral environment should be considered [11]. Salivary factors, microbial biofilms, and factors related to reconstructions are part of a unique, dynamic, and complex system that influences short- and long-term prosthetic implant therapy [12,13].

Titan-based alloys are the most utilized in dental implantology due to their stability in the human physiological environment, on account of the native amorphous oxide on their surface [14,15]. The oxide layer on the surface of the implant plays a crucial role in its stability, preventing the release of metal ions into the surrounding areas. As such, the layer of native oxides arising from the alloy's elements prevents the dissemination of corrosion processes from the surface of the biomaterial [16,17]. Discontinuities in the oxide film may occur due to the actions of active oxygen species, proteins, cells, or organic ions [18].

Implant failure may be influenced by mobility, wear, or the exposure of the implant to the oral cavity environment [19]. Peri-implantitis is a chronic pathological microbial process [20] that affects the soft tissue and surrounding bony areas of an osseointegrated dental arch implant and leads to bone resorption [21]. Peri-implantitis may favor implant rejection, increasing the accumulation of bacterial biofilm on the implant surfaces and initiating an increased number of inflammatory cells in the subepithelial conjunctive tissue [22,23]. Systemic conditions, such as HIV, may cause an increase in peri-implant infections and slightly worse results of the implant rehabilitation, which may be hindered by heavy smoking, but apparently not by oral hygiene [24–26]. Secondary implant failure related to peri-implantitis may be predicted using the plaque index (PI) and the presence of bleeding on probing (BOP) and of pocket probing depth (PPD), which have proven to be significant risk indicators [27]. The treatment of peri-implantitis may be conservatory, but surgery is an option, employing resection or regeneration [28]. Preventive measures when implanting, such as employing a partial thickness flap, may allow an adequate development of keratinized tissue around the implant, increasing implant survival [29].

Bacteria may colonize the implant's rough surface and facilitate the adherence of other colonizers, which causes a time-dependant aggression on the implant, with detrimental effects such as pitting corrosion after one month, and flexural strength decline after three months [30,31].

This in vitro study aims to use electrochemical and coupled scanning electron microscopy (SEM)–energy dispersive spectrometry (EDS) imaging to investigate the corrosion process on the surface of peri-implantitis-related rejected dental implants. To our best knowledge, a study of this kind is a novelty in the investigation field of life-time dental materials and could aid in dental bone regeneration research [32].

## 2. Materials and Methods

### 2.1. Materials

An electrochemical analysis was performed on four rejected and recovered dental implants from patients with diagnosed peri-implantitis. Written consent was obtained. The four implants are labeled Samples 1, 2, 3, and 4, respectively, and are schematically represented in Figure 1. The samples were identified in correlation with the patients' medical charts.

The manufacturers of the dental implants are not known, as implants were inserted in other dental offices and there was no data regarding the dental implants' producers from previous patients' medical data. The implants were explanted at different times. Immediately after an implant's removal, it was rinsed under a jet of water to clean the organic debris. Attached bone remnants were removed carefully, so as not to affect the surface. Implants were kept in sterile containers. Prior to the beginning of the experiment, the implants were placed in an ultrasonic bath, in ethanol, and rinsed thoroughly with water.

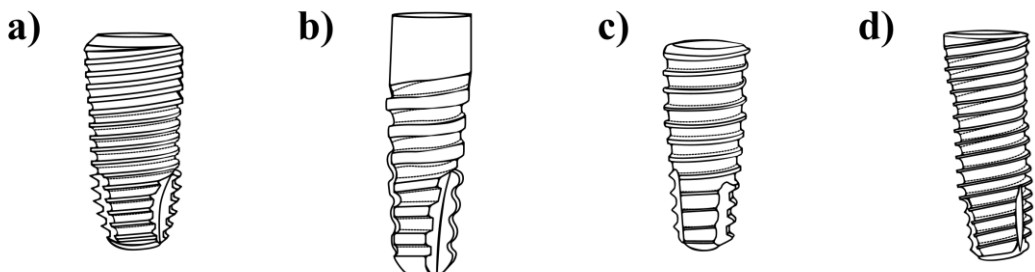

**Figure 1.** Schematic representation of the dental implants used in the study: (**a**) Sample 1, (**b**) Sample 2, (**c**) Sample 3, and (**d**) Sample 4.

A thin collar was made from light-cured resin in the cervical area of all implants in order to prevent electrolyte solution penetration at the level of attachment in the electrochemical cell. The first implant had a total length of 10 mm, a cervical diameter of 4.9 mm, a tip diameter of 2.9 mm, and 8 mm were used for the experimental investigations. The second implant had a total length of 12.3 mm, a cervical diameter of 3.7 mm, a tip diameter of 2.5 mm, and 10 mm were used for the experimental investigations. The third implant had a total length of 9.5 mm, a cervical diameter of 3.1 mm, a tip diameter of 2.3 mm, and 7 mm were used for the experimental investigations. The fourth implant had a total length of 10 mm, a cervical diameter of 3.8 mm, a tip diameter of 2.5 mm, and 8 mm were used for the experimental investigations.

The dental implants were introduced in electrochemical cells as a working electrode. Two additional electrodes were used: a reference Ag/AgCl, 3M, KCl (Metrohm) electrode and a platinum (Metrohm) counter electrode. The electrochemical stability of the samples was performed using an Autolab (Metrohm) potentiostat/galvanostat. The testing environment was a Fusayama–Meyer artificial saliva 0.4 g/L NaCl, 0.4 g/L KCl, 0.795 g/L $CaCl_2 \cdot 2H_2O$, 0.69 g/L $NaH_2PO_4 \cdot H_2O$, 0.05 $Na_2S \cdot 9H_2O$, and 1 g/L urea, 6.5 pH [33].

### 2.2. Methods

Electrochemical analysis of the implants was performed with open circuit potential (OCP) measurements, Tafel analysis, and electrochemical impedance spectroscopy (EIS), as well as SEM coupled with an electron probe micro-analyzer for EDS. The immersion period of the implants was 168 h (one week). Measurements were performed at $t = 0$ (initial immersion of the implants into the testing environment), 24, 72, and 168 h.

### 2.2.1. Open Circuit Potential Measurements

The OCP measurements were performed using the Autolab PGSTAT 302N (Metrohm, Herisau, Switzerland) potentiostat/galvanostat.

### 2.2.2. Electrochemical Impedance Spectroscopy

The EIS measurements were performed using the Autolab PGSTAT 302N (Metrohm, Herisau, Switzerland) potentiostat/galvanostat. The data was analyzed using the Nova 1.11 software. The measurements were performed in the 10 kHz–0.1 Hz range at an amplitude of 20 mV.

### 2.2.3. Tafel Analysis

Tafel determinations were obtained using the Autolab PGSTAT 302N (Metrohm, Herisau, Switzerland) potentiostat/galvanostat. Linear potentiostat polarization was obtained by applying a ±150 mV current compared to the OCP at a scanning rate of 2 mV/s. The data was analyzed using the Nova 1.11 software.

### 2.2.4. Scanning Electron Microscopy

The SEM images were obtained using a Quanta 650FEG scanning electron microscope from Thermo Fisher Scientific (Waltham, MA, USA) paired with an Octane Silicon Drift Detector EDS sensor (EDAX, AMETEK, Inc., Berwyn, PA, USA) using the following parameters: acceleration 10 keV, spot size 4 nm, working distance 14 mm, dwell time 1 μs, and a pressure of 7.5 mPa.

### 2.2.5. Statistical Analysis

Corrosion rates and polarization resistance were summarized as means, standard deviations (SDs), and medians, per sample. Non-parametric tests were used for intergroup comparisons. The level of significance was set at 0.05. We performed statistical analyses using Stata/IC 16 (StataCorp, College Station, TX, USA).

## 3. Results

### 3.1. Dental Implants

The first implant (Sample 1) was obtained from a 64-year-old male patient. The rejected implant was identified on the orthopantomography (OPG) in position 3.5 (Figure 2, arrow). The implant was placed approximately five years ago and the prosthesis was a porcelain fused to metal (PFM) crown. At initial appointment, the patient complained of tenderness and bleeding on brushing at the implant sites. The cervical part of the implant was uncovered in the oral cavity. The clinical examination revealed implant mobility, gingival retraction, local inflammation, and poor oral hygiene. The patient interview uncovered he was a smoker of approximately 20 cigarettes per day and suffered from untreated arterial hypertension. The implant was removed and the site was allowed to heal for four months, then another implant was inserted at the same site.

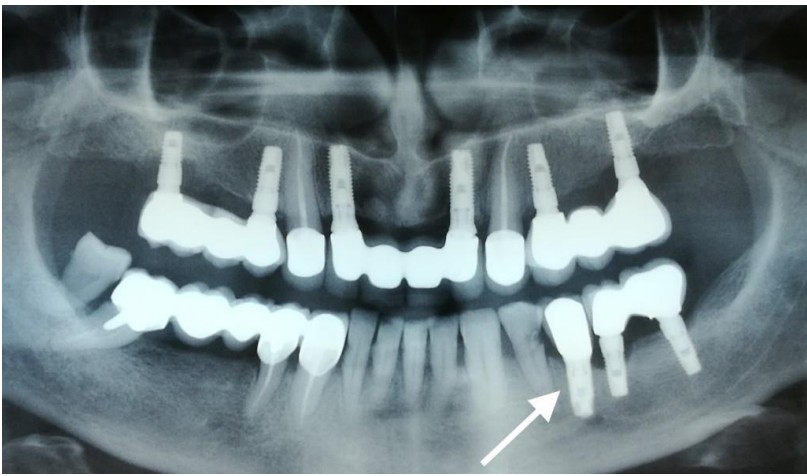

**Figure 2.** OPG radiography revealing peri-implantitis in quadrants 1, 2, and 3. The implants inserted in positions 1.2, 1.4, 1.6, 2.2, 2.4, 2.6, 3.5, 3.6, and 3.7 show generalized vertical and horizontal bone loss. The investigated implant is marked with an arrow.

The second implant (Sample 2) was recovered from a 55-year-old male patient and was inserted in position 3.4 (Figure 3, arrow). The rehabilitation included a PFM crown. The patient presented with pain and mobility of dental implants in quadrant 3. Following the clinical examination, poor oral hygiene was detected by the presence of calculus and soft plaque deposits. The implants in quadrant 3 showed gingival retraction and exposure of the first grooves of the cervical segment to the oral cavity. The patient interview revealed he was a heavy smoker (of around 30 cigarettes per day) and suffered from arterial hypertension treated with prescribed medication.

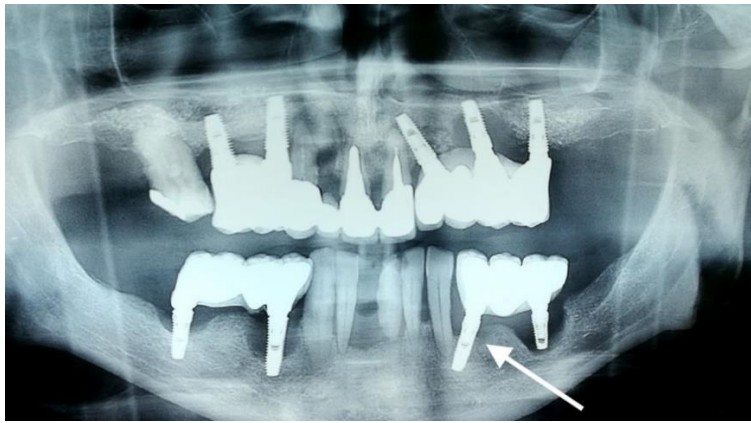

**Figure 3.** OPG radiography revealing peri-implantitis-affected areas. Quadrants 3 and 4 show bone loss extended to the apical segments of the implants, while quadrants 1 and 2 show a goblet-shaped bone loss. The studied implant is marked with an arrow.

The third rejected implant (Sample 3) was recovered from position 4.5 of a 49-year-old woman, and is marked with an arrow on the dental radiography (Figure 4, arrow). Both implants in the lower right jaw were placed under local anesthesia in one session. After four years, the implants and the bridge reconstruction failed. The cervical segment of the implant was uncovered in the oral cavity. The patient reported pain in the areas of implantation, accompanied by mobility, gingival retraction, and inflammation. The interview and clinical examination revealed average oral hygiene, smoker status (20 cigarettes/day), and diabetes mellitus controlled by diet and prescription medication.

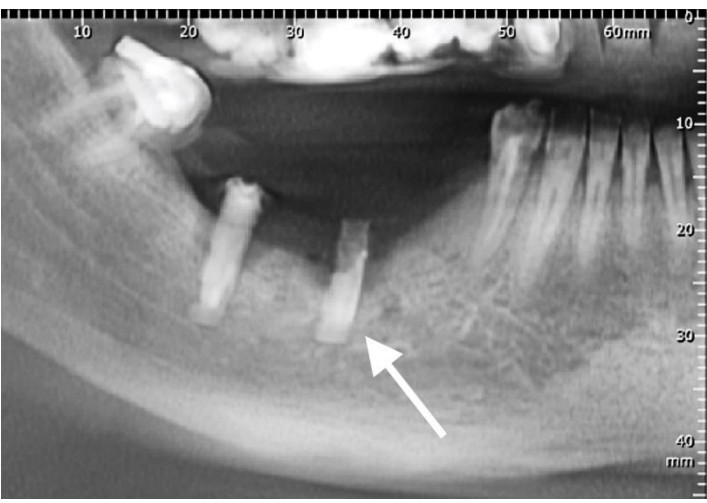

**Figure 4.** Radiography centered in quadrant 4 displaying incorrect positioning of inserted implants. Peri-implantitis with bone loss and exposure of the cervical segment of the implant are noted. The studied implant is marked with an arrow.

The fourth implant (Sample 4) was recovered from quadrant 4, position 4.5 of a 66-year-old male patient (Figure 5, arrow). The patient presented with pain in the area of dental implants, as well as mobility, gingival retraction, and inflammation. Poor oral health was noted at the clinical examination, with calculus deposits and food remnants. The patient was a smoker, suffering from diabetes mellitus treated with diet and medication. The implant exposed its cervical segment to the oral cavity due to bone loss. After a panoramic scan, a total of seven implants were visualized. Intraorally, there was a removable partial overdenture and a metal bar infrastructure on three of the inferior implants. Four implants appeared to be functionally viable. The lower right implants had

severe peri-implant tissue involvement and substantial bone loss. The distal implant on the left side also presented peri-implantitis.

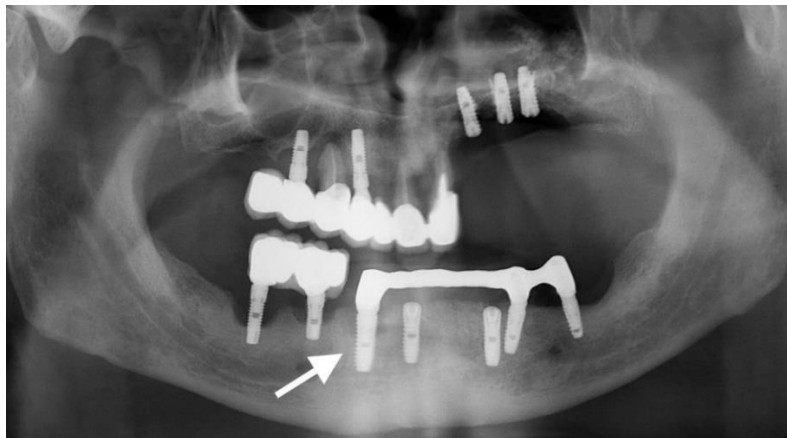

**Figure 5.** OPG radiography. Peri-implantitis in quadrants 3 and 4 through bone loss identified up to the apical segment for the implant in position 4.3 and generalized horizontal bone loss around the other implants.

The peri-implantitis was not addressed conservatively (local debridement and implant scaling) due to its severe status. The affected implants were explanted using rotary instruments (trephine drills) and a specific implant removal kit.

### 3.2. Electrochemical Testing

#### 3.2.1. Open Circuit Potential Measurements

Figure 6 presents a diagram of the potential time variation. Immediately after immersion in the electrolyte solution, Sample 3 demonstrated the most positive potential (0.088 V) at $t = 0$ h, while Sample 4 showed the most electronegative potential ($-0.509$ V). Samples 2 and 3 displayed a similarly electronegative potential at $t = 0$ h. At 24 h, a potential declining trendline was evidenced in the four samples, with the most marked drop in Sample 3, reaching a value of $-0.399$ V, while the most electronegative value was recorded for Sample 4, with $-0.528$ V. All samples showed a decrease in potential, denoting a depassivation of the metallic biomaterial surface.

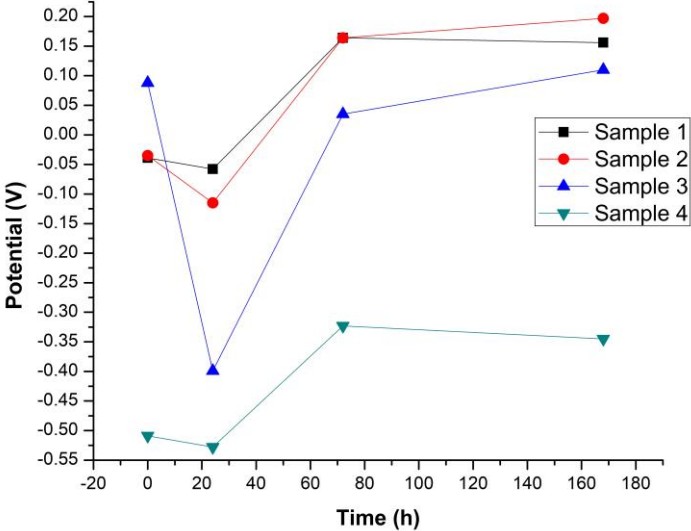

**Figure 6.** OCP diagrams for Samples 1–4, followed for 168 h.

Three days from the immersion, Samples 1 and 2 reached the highest value (0.164 V) while Sample 4 was the only one to remain electronegative (−0.323 V). Nevertheless, a trend of increase in the potential of all samples was noted, indicating the tendency of depassivation of the native oxides layer on the surface of the implants. Between 72 and 168 h, an increase in the measured potential was noted for Samples 2 and 3, indicating the constitution of a protective oxide layer on the surface. Conversely, for Samples 1 and 4, the potential showed a decrease, which can occur in the case of a discontinuity in the oxide layer.

### 3.2.2. Tafel Analysis

Tafel diagrams for voltammetry slope variations are displayed in Figure 7, corrosion rates are shown in Figure 8, and the main corrosion parameters for the four samples are presented in Table 1. The summary statistics for corrosion rates showed for Sample 1—mean 0.025 (SD 0.05; median 0.002), Sample 2—mean 0.01 (SD 0.02; median 0.0004), Sample 3—mean 0.0008 (SD 0.001; median 0.0001), and Sample 4—mean 0.029 (SD 0.03; median 0.02). The corrosion rate was higher in Sample 4 than Sample 3 ($p = 0.02$, Mann–Whitney test). The polarization resistance measurements can be summarized as follows: Sample 1—mean 290 (SD 317; median 245.72), Sample 2—mean 434.75 (SD 392.11, median 390.5), Sample 3—mean 885.34 (SD 735.24, median 732.49), and Sample 4—mean 29.68 (SD 18.21, median 24.55). No significant differences were recorded between samples ($p = 0.134$, Kruskal–Wallis test).

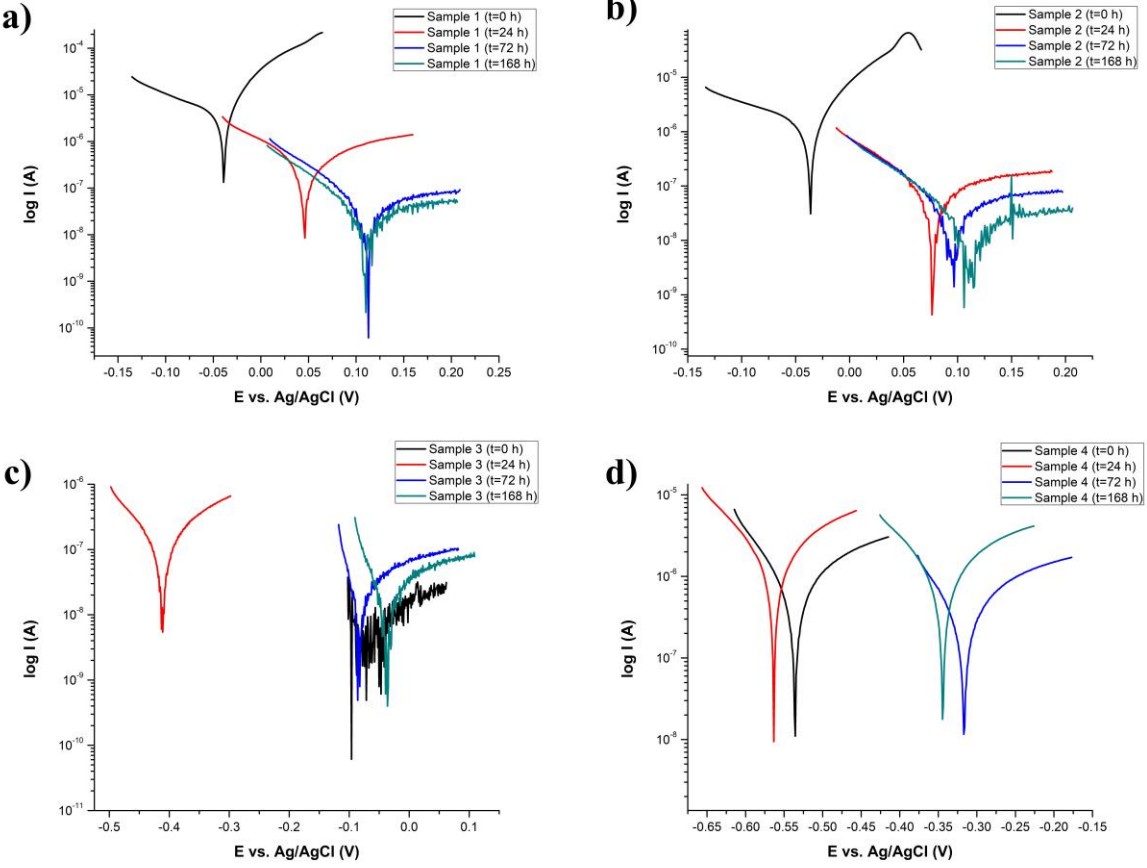

**Figure 7.** Tafel diagrams for Samples 1 (**a**), 2 (**b**), 3 (**c**), and 4 (**d**) at different electrolyte solution immersion times.

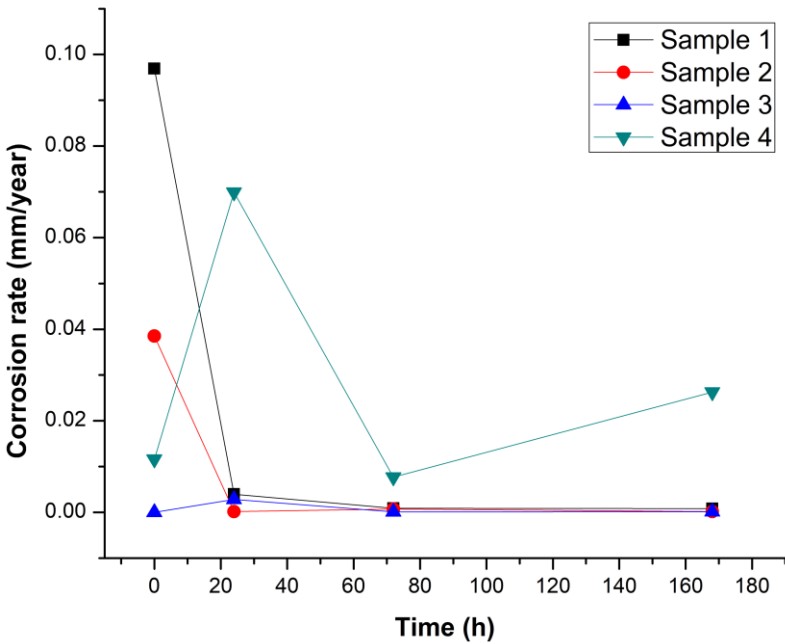

**Figure 8.** Corrosion rate curves for Samples 1–4 throughout 168 h of immersion.

**Table 1.** Tafel parameter values—corrosion potential, current density, current intensity, and polarization resistance for Samples 1–4 over 168 h.

| Sample (Time) | $E_{corr}$ (V) | $J_{corr}$ ($\mu$A/cm$^2$) | $I_{corr}$ ($\mu$A) | Corrosion Rate (mm/year) | Polarization Resistance (k$\Omega$) |
|---|---|---|---|---|---|
| Sample 1 ($t$ = 0 h) | −0.0386 | 11.379 | 12.630 | 0.096932 | 2.614 |
| Sample 1 ($t$ = 24 h) | 0.0461 | 0.462 | 0.513 | 0.003940 | 53.767 |
| Sample 1 ($t$ = 72 h) | 0.1091 | 0.102 | 0.113 | 0.000873 | 437.67 |
| Sample 1 ($t$ = 168 h) | 0.1086 | 0.094 | 0.105 | 0.000808 | 665.98 |
| Sample 2 ($t$ = 0 h) | −0.0361 | 4.520 | 4.703 | 0.038507 | 8.67 |
| Sample 2 ($t$ = 24 h) | 0.0783 | 0.023 | 0.021 | 0.000173 | 316.34 |
| Sample 2 ($t$ = 72 h) | 0.0947 | 0.083 | 0.086 | 0.000709 | 464.66 |
| Sample 2 ($t$ = 168 h) | 0.1053 | 0.019 | 0.019 | 0.000162 | 949.34 |
| Sample 3 ($t$ = 0 h) | −0.0926 | 0.00015 | 0.001 | 0.000013 | 1907.70 |
| Sample 3 ($t$ = 24 h) | −0.4110 | 0.331 | 0.407 | 0.002821 | 168.70 |
| Sample 3 ($t$ = 72 h) | −0.0837 | 0.013 | 0.016 | 0.000115 | 642.49 |
| Sample 3 ($t$ = 168 h) | −0.0392 | 0.017 | 0.021 | 0.000149 | 822.48 |
| Sample 4 ($t$ = 0 h) | −0.5354 | 1.361 | 1.522 | 0.011599 | 27.75 |
| Sample 4 ($t$ = 24 h) | −0.5635 | 8.208 | 9.176 | 0.069919 | 13.96 |
| Sample 4 ($t$ = 72 h) | −0.3166 | 0.902 | 1.008 | 0.007683 | 55.66 |
| Sample 4 ($t$ = 168 h) | −0.3439 | 3.082 | 3.446 | 0.026257 | 21.36 |

Voltammetry curves in Figure 7 showed a trend of increase of the electrochemical stability through electropositive shifts for the entire observed period for Samples 1 and 2 (Figure 7a,b, respectively). Similar findings were registered for Sample 3 (Figure 7c) with the exception of the initial measurement ($t$ = 0 h), where the signal-to-noise ratio was very low. Sample 4 (Figure 7d) displayed similar characteristics of decreased stability with electronegative potentials in the first 24 h followed by increased stability with increasingly electropositive potentials.

The first 24 h seemed to be defining moments for the evolution of the electrochemical characteristics in all samples. The voltammetry slopes shift in Sample 1 suggested that the material tends to stabilize towards electropositive potentials at 72 and 168 h. The same trend of stability due to implant material surface passivation through a stable passive oxide layer was noted for Sample 2 where the corrosion slopes shifted towards electropositive values. Better stability was observed at 24 and 72 h by correlated corrosion potential values of 0.078 and 0.094 V, respectively. Sample 3 showed a shift towards a more electropositive potential of the corrosion slopes at 72 and 168 h, in agreement with the higher corrosion potential values of −0.084 V and −0.039 V recorded in Table 1. Corrosion curves for Sample 4 (Figure 7d)

indicated a positioning of the voltammetry slopes in the electronegativity spectrum, the respective curves for 72 and 168 h demonstrating the highest sample stability throughout the experiment.

Corrosion rate variation is represented in Figure 8. The lowest value for Sample 4 (0.011 mm/year) and highest value for Sample 1 (0.096 mm/year) were recorded at $t = 0$ h. At 24 h the corrosion rate for Sample 1 decreased to 0.003 mm/year, resembling that of Sample 2 (0.00017 mm/year) and Sample 3 (0.002 mm/year). Sample 4's corrosion rate varied the most, reaching 0.069 mm/year. After 72 h of immersion, the corrosion rate of Samples 1 through 3 showed a declining trend while Sample 4 registered a marked decrease, reaching 0.007 mm/year. At 168 h, Sample 4 recorded an increase in corrosion rate, reaching 0.026 mm/year, while Samples 1 through 3 registered a plateau, denoting a stability tendency.

Polarization resistance is defined as the transition resistance between electrodes and the electrolyte. An increased resistance to the current flow in a voltaic cell is caused by the chemical reaction to the electrodes. Polarization leads to a decrease of electrical potential in the electrochemical cell. Polarization resistance in the four samples showed variations throughout the 168 h of immersion. At $t = 0$ h, the implants demonstrated similar values. However, starting at $t = 24$ h up to $t = 168$ h, an increase of polarization resistance was noted for Samples 1, 2, and 3, which related to the values of current intensity and corrosion rates in this period. Sample 4 demonstrated a slight decrease towards 13.96 kΩ at $t = 24$ h, a slight increase (55.66 kΩ) at 72 h, followed by another decrease (21.36 kΩ) at 168 h. The registered values for Tafel parameters in Sample 3 were very low at immersion time ($t = 0$ h). This may indicate a quick surface passivation process in Sample 3. EIS measurements in relative electrochemical passivation were used as ancillary electrochemical investigation methods to obtain relevant information for Sample 3 at immersion time.

Table 1 shows the values of the Tafel parameters: corrosion potential, current density, current intensity, and polarization resistance for Samples 1–4 over 168 h of immersion in Fusayama–Meyer artificial saliva.

### 3.2.3. Electrochemical Impedance Spectroscopy

EIS measurements were performed for each sample over the 168 h of the experiment. Equivalent electric circuits for each sample were proposed and fitted at $t = 0$ h (at immersion), $t = 24$ h, $t = 72$ h, and $t = 168$ h (endpoint). Bode Modulus (impedance module vs. frequency) and Nyquist (imaginary vs. real impedance) diagrams were obtained and interpreted from the experimental and fitted data derived from the EIS measurements.

Bode Modulus diagrams for Samples 1–4 are presented in Figure 9. Three frequency areas were defined for these results: (i) low-frequency area (0.1–10 Hz), (ii) mid-frequency area (10–100 Hz), and (iii) a high-frequency area (above 100 Hz).

At low frequencies, at the beginning of the experiment ($t = 0$ h) Samples 1 and 2 showed the lowest average impedance value of 800 Ω for Sample 1 and 1500 Ω for Sample 2, respectively. At 24 h of immersion, the resistances of these samples rose to around 11,000 Ω at 0.1 Hz for both samples. Sample 3 (Figure 9c) displayed stability at low frequencies where the curves of the four measurements overlapped. For Samples 3 and 4 the measured impedance held at values around 11,500 Ω and 10,000 Ω, respectively, at 0.1 Hz for all immersion times.

For all samples, the mid-frequency area was a transitional zone to a relatively constant region of impedance values recorded at high frequencies. At high frequencies, Sample 1 (Figure 9a) displayed a tendency towards stability. Sample 2 (Figure 9b) showed an increase in total impedance for the measurements at $t = 72$ h. Conversely, Sample 3 (Figure 9c) demonstrated an increase in total impedance at $t = 72$ h, followed by $t = 24$ h, $t = 0$ h, and $t = 168$ h. Sample 4 (Figure 9d) recorded minimal variation in impedances in all frequency domains (low, mid, and high).

The low impedances of Samples 1 and 2 observed at the initial moment ($t = 0$ h) in the Bode representation were due to low resistances, which may be observed in Figure 10. The Nyquist diagrams represent the imaginary part of impedance, -Im Z″, depending on the real part of the impedance, Re Z′,

at various immersion times in the artificial saliva solution for Samples 1–4 (Figure 10a–d, respectively). These diagrams revealed that Sample 3 demonstrated the most capacitive behavior out of all samples, evidenced by the vicinity of the curves and the imaginary part of impedance, -Im Z. Sample 1 showed the most resistive behavior out of all samples.

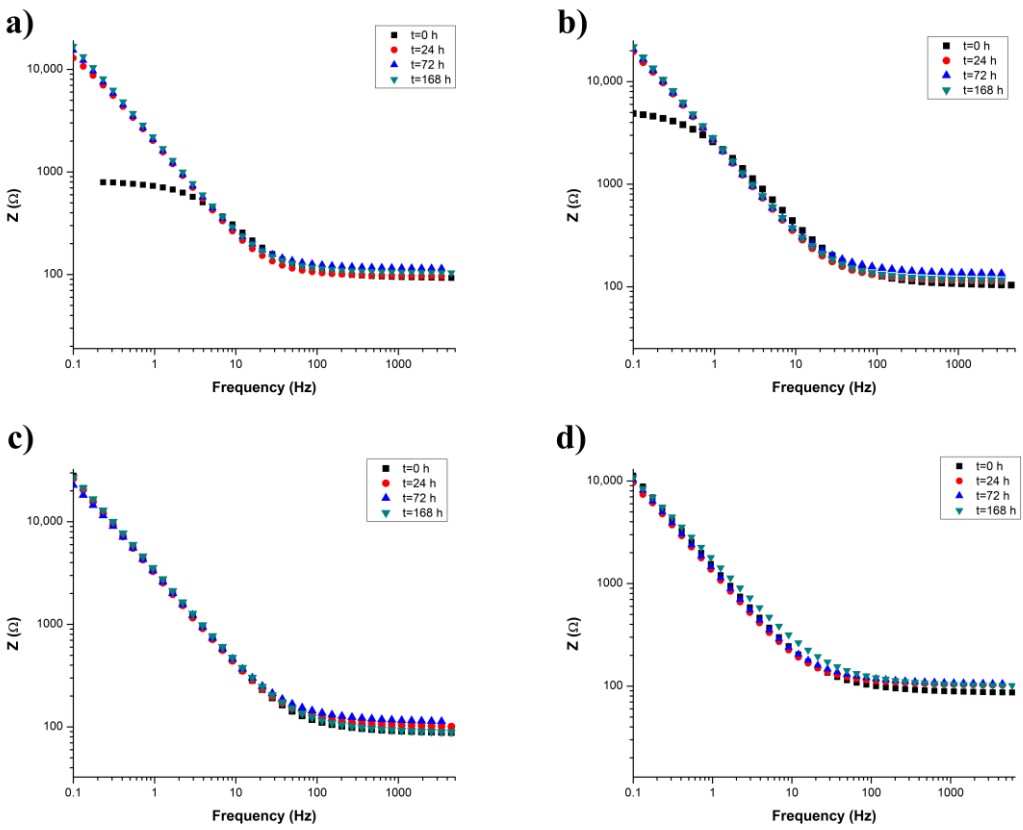

**Figure 9.** Bode Modulus diagrams for (**a**) Samples 1, (**b**) 2, (**c**) 3, and (**d**) 4.

An equivalent Randles circuit was proposed for each sample, made up of the resistance of the electrolyte solution connected in series with the pseudocapacitance of the double-electric layer as a constant phase element (CPEi) and a resistance attributed to the dental implant (Ri), both connected in parallel. Table 2 presents the values of the electrical circuits' parameters for Samples 1 through 4 at *t* = 0, 24, 72, and 168 h. The term Y0 of the constant phase element defines the capacitance of the double electrochemical layer, while the n factor defines the power of the constant phase element, with values ranging from 0 (ideal resistor) to 1 (ideal capacitor).

The proposed and fitted values for the equivalent circuits are presented in Table 2. The n factor values for Samples 1–4 reveal their capacitive character over the 168 h of immersion. Sample 3 is the implant with the highest initial resistance (one order of magnitude above the other samples at *t* = 0), as was proposed by the Tafel data regarding the quick surface passivation. Samples 1 and 2 demonstrated an increase of resistance values attributed to the implanted material, reaching 56,600 Ω and 75,000 Ω, respectively, at *t* = 168 h. Conversely, Sample 3 showed a decline in material resistance values towards *t* = 72 h, followed by a slight increase at *t* = 168 h. Sample 4, the least stable as suggested by the Tafel analysis, presented the lowest values of implant resistance, decreasing steadily from 115,000 Ω at *t* = 0 to 29,000 Ω at *t* = 168 h.

**Table 2.** Electrical circuit parameters for Samples 1–4 (at $t$ = 0, 24, 72, and 168 h).

| Sample (Time) | Fusayama–Meyer Saliva | | Dental Implant | | $\chi^2$ |
| | Rsol (Ω) | Ri (Ω) | CPEi | | |
| | | | $n$ | Y0 (µMho) | |
|---|---|---|---|---|---|
| Sample 1 ($t$ = 0 h) | 94.1 | 719 | 0.881 | 98.7 | 0.0109 |
| Sample 1 ($t$ = 24 h) | 95.7 | 26,300 | 0.934 | 94.5 | 0.0193 |
| Sample 1 ($t$ = 72 h) | 118 | 50,000 | 0.933 | 91.9 | 0.0142 |
| Sample 1 ($t$ = 168 h) | 107 | 56,600 | 0.936 | 85.5 | 0.0147 |
| Sample 2 ($t$ = 0 h) | 104 | 5150 | 0.884 | 66.6 | 0.0375 |
| Sample 2 ($t$ = 24 h) | 114 | 55,900 | 0.927 | 70.9 | 0.0359 |
| Sample 2 ($t$ = 72 h) | 134 | 72,000 | 0.921 | 70.7 | 0.0960 |
| Sample 2 ($t$ = 168 h) | 116 | 75,000 | 0.927 | 67.2 | 0.0554 |
| Sample 3 ($t$ = 0 h) | 88 | 1,000,000 | 0.913 | 57.3 | 0.1061 |
| Sample 3 ($t$ = 24 h) | 101 | 295,000 | 0.909 | 59.6 | 0.0610 |
| Sample 3 ($t$ = 72 h) | 113 | 200,000 | 0.889 | 62.4 | 0.0500 |
| Sample 3 ($t$ = 168 h) | 89 | 277,000 | 0.9 | 56.1 | 0.0804 |
| Sample 4 ($t$ = 0 h) | 86.7 | 115,000 | 0.873 | 136 | 0.0558 |
| Sample 4 ($t$ = 24 h) | 105 | 36,000 | 0.891 | 148 | 0.0215 |
| Sample 4 ($t$ = 72 h) | 110 | 66,500 | 0.882 | 143 | 0.0270 |
| Sample 4 ($t$ = 168 h) | 101 | 29,000 | 0.829 | 126 | 0.0500 |

Rsol: the resistance for the artificial saliva solution; Ri: the resistance for the dental implant; CPEi: the pseudocapacitance of the implant's double-electric layer as a constant phase element; Y0 denotes the pseudocapacitance value.

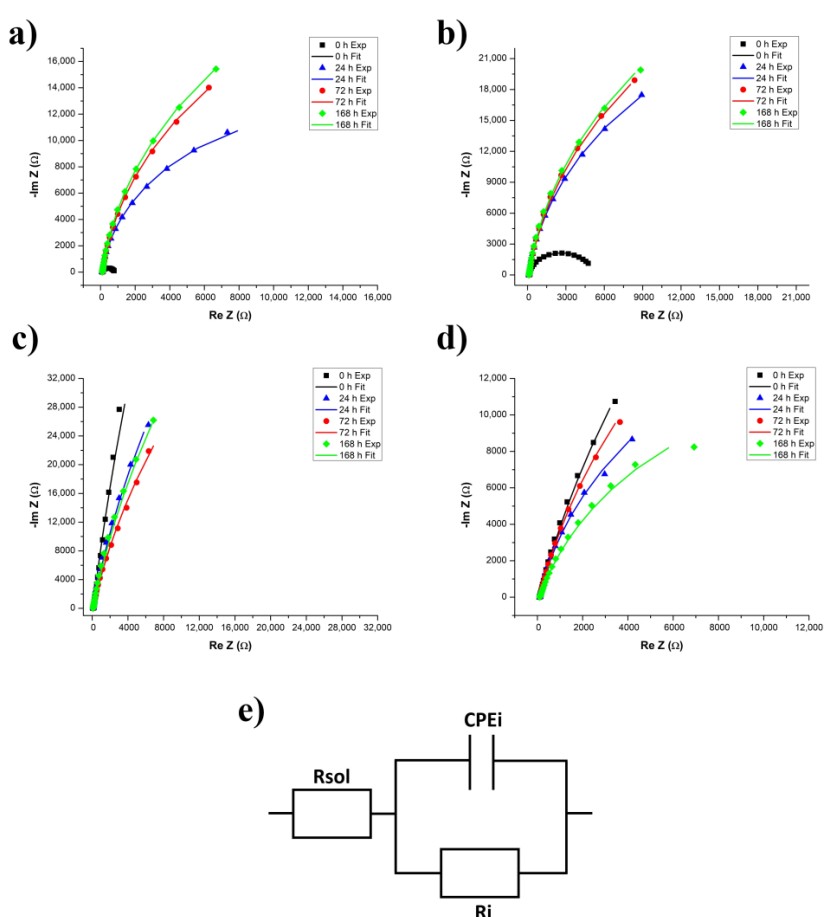

**Figure 10.** The electrochemical impedance spectroscopy measurements in artificial saliva at various numbers of immersion times and the fitted data for (**a**) Sample 1, (**b**) Sample 2, (**c**) Sample 3, (**d**) Sample 4, and (**e**) the EIS-proposed circuit used to fit these experiments. The symbols represent the experimental data (Exp), while the continuous lines represent the fitted (Fit) results.

### 3.3. Scanning Electron Microscopy Coupled with an Energy Dispersive Spectroscopy Probe

The SEM images were obtained prior to electrochemical measurements. Figure 11 shows the different types of implants, with their respective diameters and groove increments. The biggest diameter was recorded in Sample 2, while the smallest was in Sample 3. Sample 2 (Figure 11b) displayed bone tissue deposits that were subsequently mechanically removed with Teflon instruments and ethanol ultrasonic cleaning prior to the electrochemical determinations in order to prevent measurement bias by surface alterations. Samples 1, 3, and 4 displayed pitting corrosion areas on the surface, more prominent in Samples 3 and 4. Figure 12 shows the different roughness of the implants. Surface roughness in implant materials favors and stimulates osteoblast cells to osseointegrate the implant in the dental arch.

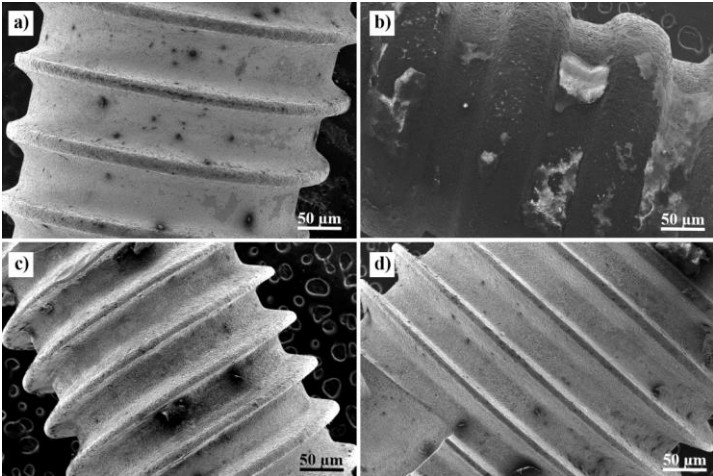

**Figure 11.** SEM micrograph (100× magnification) showing implant surfaces for (**a**) Samples 1, (**b**) 2, (**c**) 3, and (**d**) 4.

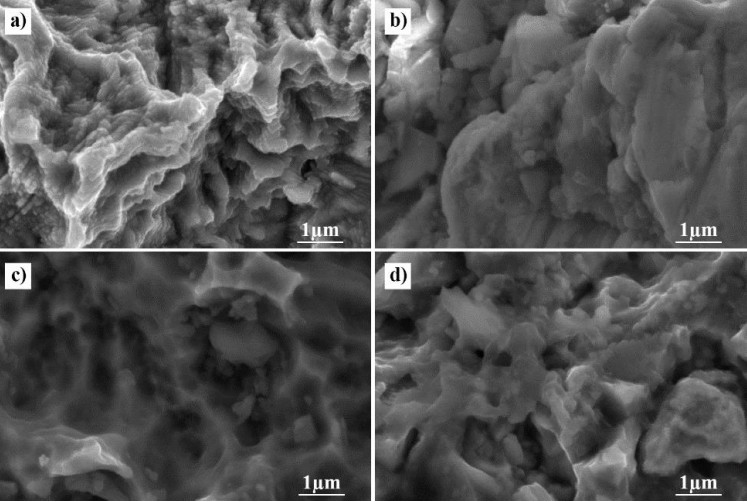

**Figure 12.** SEM micrograph (10,000× magnification) showing implant morphology for (**a**) Samples 1, (**b**) 2, (**c**) 3, and (**d**) 4.

The EDS spectra (Figure 13) showed the highest intensity for Ti in all implants. Aluminum was observed in increasing proportions in Samples 1, 3, 4, and 2, in this order. However, an important aspect identified in all EDS spectra was the presence of the Cl element, which is associated with pitting corrosion. Table 3 presents the elemental EDS composition in mass percentages for the four analyzed implants. Because the vanadium band was notably close to the noise level (as also observed in data for

similar materials) [34], it could not be reliably detected. Ti held the highest amplitude, followed by Al. The presence of P and Ca elements was determined by the presence of hydroxyapatite from bone tissue remnant deposits on the surface of the alloy, subsequently removed through mechanical and ultrasonic cleaning prior to electrochemical analysis.

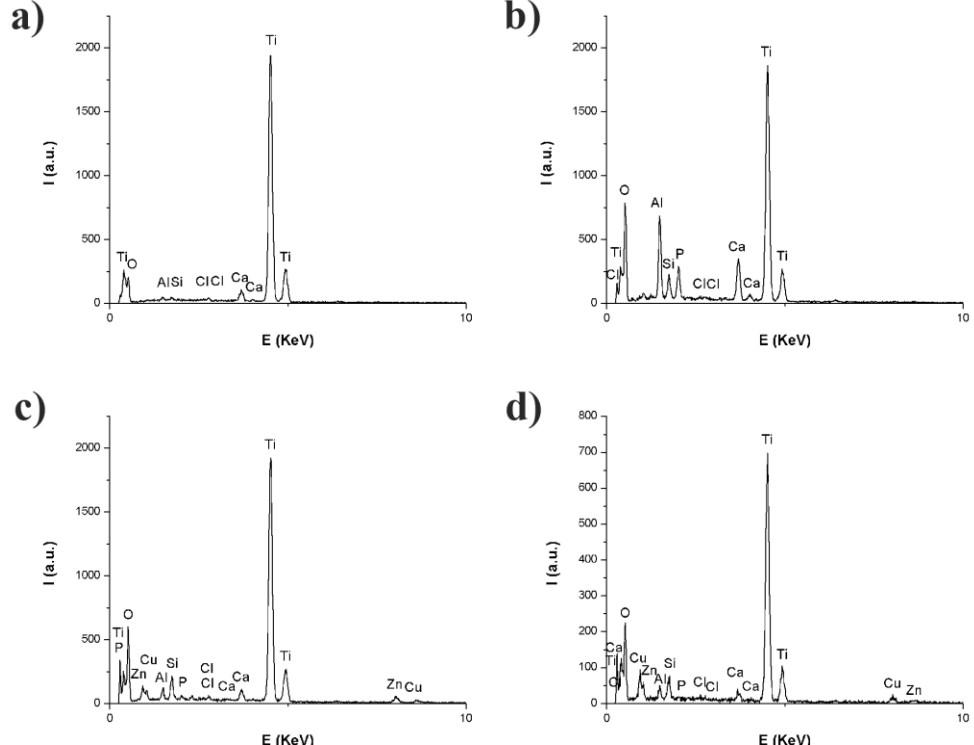

**Figure 13.** Relative composition of the atomic elements determined through EDS for the implanted materials: (**a**) Samples 1, (**b**) 2, (**c**) 3, and (**d**) 4.

**Table 3.** Elemental EDS composition in mass percentages (%) for the studied implants.

| Sample | Element % | | | | | | | | | |
|---|---|---|---|---|---|---|---|---|---|---|
| | O | Al | Si | P | S | Cl | Ca | Ti | Cu | Zn |
| Sample 1 | 4.44 | 0.30 | 0.16 | 0.00 | 0.00 | 0.14 | 2.27 | 54.6 | 0.00 | 0.00 |
| Sample 2 | 58.7 | 6.59 | 1.56 | 2.43 | 0.00 | 0.15 | 4.21 | 26.3 | 0.00 | 0.00 |
| Sample 3 | 57.5 | 1.17 | 1.99 | 0.37 | 0.30 | 0.23 | 1.60 | 33.9 | 1.89 | 0.93 |
| Sample 4 | 58.9 | 1.37 | 1.95 | 0.30 | 0.18 | 0.30 | 1.21 | 33.0 | 1.64 | 1.10 |

## 4. Discussion

The present study contributes to the investigation of corrosion in rejected dental implants in patients with peri-implantitis pathology. The colonization of the peri-implant sulcus by Gram-negative anaerobes alongside other factors, such as poorly controlled diabetes, smoking, implant design, and mechanical stress, creates an inflammatory environment facilitating loss of bony support and ultimately leading to implant failure [35]. The bacterial profile seems to correlate with the degree of inflammation and the prognosis of the implant, however, the surface structure of the implant is also an important factor, due to the attachment affinity of some bacteria to specific implant surface types [36]. The inconsistently reported prevalence of peri-implantitis seems to confirm that it is a complex multifactorial process, and the correct identification of bacterial pathogens to peri-implantitis may help limit the disease severity [37]. Patients' radiographs showed direct exposure of part of the implants to the oral environment following vertical and horizontal bone resorption at the insertion sites. This exposure set the implants in contact with human saliva, a natural fluid. Saliva acts as an

electrolyte solution in an environment predisposed to oral galvanism and corrosion processes, while also acting directly on the elements composing the implant.

Patients' radiographs, OCP measurements, main corrosion parameters, EIS data, SEM images, and EDS spectra were interpreted and correlated to offer a better understanding of the corrosion processes with unwanted effects on osseointegration and implant life duration. Closed-circuit potential or resting potential is a passive method used to measure the electrochemical potential of the electrolyte solution. Following the OCP measurements, a potential variation of the four samples during the 168 h of immersion was recorded. These alternations affect the protective native oxide film on the surface of the implant. Decreases in potential represent film discontinuities, indicating depassivation cycles, while increases in potential indicate passivation processes through oxide layer restoration. The succession between passivation and depassivation indicates the samples' tendency towards instability throughout the experimental period. Sample 2 was observed to present the best tendency towards stability, with the highest potential values, while Sample 4 presented the lowest values, and was the only one displaying persistent electronegativity. Unlike the Tafel analysis, where a corrosion-triggering potential is applied, OCP measurements are non-invasive and record the resting electrochemical potential of the electrolyte solution.

The Tafel analysis parameters are interdependent, thus a high corrosion rate is associated with a higher current intensity and a lower polarization resistance. The first three samples show a higher tendency towards stability, observed after 24 h of immersion by the decrease in corrosion rates and current intensities, and a corresponding increase in polarization resistance. This phenomenon may be explained by the formation of a protective native oxide layer on the implant's surfaces that leads to a decrease in ion release rate [38]. These results allow for a correlation with EIS data [39], the first three samples demonstrating the highest values of resistances and more capacitive phase angle values, while the last sample showed decreased values of both Tafel parameters and the proposed and fitted electric circuit resistances. Sample 4 displayed the highest corrosion rate over time and showed important variations in charge transfer resistance, demonstrating the lowest values at the end of the measurement period.

EIS results showed the highest implant resistance for Samples 3 and 4 at immersion time ($t = 0$) in interface metastability conditions. The best corrosion resistance was noted for glucose concentrations of 10 mmol/L [40], which are commonly identified in patients with diabetes mellitus. Samples 3 and 4 were recovered from patients suffering from diabetes mellitus, confirming the effects of glucose on implant resistance. In contrast, diabetes-free patients yield lower resistances, as noted in Samples 1 and 2. Furthermore, a review of relevant literature showed that patients with poorly controlled diabetes also demonstrate a weaker implant osseointegration [41].

The abutment-fixture connection geometry was an internal hexagon type. The implant manufacturer delivers prefabricated abutments, compatible with the implant system, from the same alloy as the implant. Given its internal placement, the nature of the same alloy, the abutment should not influence corrosion that occurs at the external surface of the implant.

SEM images showed pitting corrosion processes, which are specific to metallic materials with surfaces protected by native oxide layers. In the case of titan-based alloys, this layer arises from contact with oxygen, creating an adherent $TiO_2$ film. In the presence of chloride ions, the oxide film is damaged, exposing fragments of the alloy surface, contributing to the initiation of corrosion. Patients' radiographs demonstrated the uncovered implants, partly exposed to the oral environment following bone resorption. Chloride was present in the natural saliva composition, and the highest concentrations were recorded in the early morning [42]. Titanium chlorides are formed in the areas of corrosion and have the tendency to hydrolyze and lead to pitting corrosion. The presence of chloride was demonstrated as peaks on the EDS spectra.

Ti grades from 1 to 4 represent Ti samples with decreasing purity from 1 to 4. Ti grades of 5 and above are representative for Ti alloys [43]. Based on the Ti/O ratio, it can be inferred that Sample 1 was Grade 1 titanium while the others were lower-purity higher-grade titanium implants. Sample 2

was most likely Ti-6Al-4V, while Samples 3 and 4 were most likely Grade 4 Ti. Interestingly, due to the higher oxygen content for Grades 4 and 5, these samples had much higher implant resistances, as EIS circuit fit showed. The implant resistance (Ri) for Sample 1 was higher than the other samples by at least one order of magnitude. Samples 1 and 2 did not show any copper, zinc, or sulfur in their elemental composition, following the EDS determination, compared to Samples 3 and 4. This may explain the low compatibility and lack of osseointegration that played a significant role in the implant rejection. Sulfur is an important component of proteins and is found in high quantities in the oral cavity in the composition of filaggrin [44]. Filaggrin (filament aggregating protein) is a filament protein connected to keratin fibers in the epithelial cells. In the epithelial tissue, these structures are found in keratohyalin granules in the granulous layers [45]. This protein is essential in the homeostasis of the epithelial tissue. In the corneous layer, filaggrin monomers are part of the skin barrier structures. Alternatively, these proteins may interact with keratin intermediary filaments. The impact of the keratinized gum on dental implants has long been debated and is a subject of controversy, however, most studies underline the importance of an adequate keratinization area around implants [46,47].

Some studies have shown an association between the lack of keratinized tissue and slight bone loss [48], with a higher accumulation of bacterial plaque and increased soft tissue retraction [49]. Alongside these clinical signs, an increased bleeding on probing index was recorded, noting a significant increase in gingival inflammation [50]. The discontinuity of the oxide film exposes a fragment of the alloy surface to the external environment, leading to a release of metallic ions and the initiation of alloy corrosion. The intensity of the galvanic effect is influenced by the potential difference between the metals that trigger this process [51]. In dental implantation, the exposure is to the oral environment and the presence of saliva. Salivary ions, such as chloride, sodium, calcium, and potassium, but also proteins, enzymes, and microorganisms of the oral biofilm, may interact with and influence the corrosion process [52,53].

## 5. Conclusions

The OCP measurements of the four implants indicated passivation–depassivation cycles, suggesting thickening and discontinuities of the passive native oxide film. A variable electrochemical behavior throughout the measurement period was identified by the Tafel analysis, suggesting a tendency towards instability. By proposing and fitting equivalent electric circuits, the EIS data indicated a better performance of the first three implants while the fourth showed the lowest values for charge-transfer resistance. Pitting corrosions on the implant's surfaces were demonstrated by SEM imaging, and confirmed by the EDS spectra, which identified chloride, an element associated with this process. Direct exposure to the oral environment favors the initiation of corrosion, which has a negative influence on the stability and osseointegration of dental implants.

**Author Contributions:** Conceptualization, I.B., M.A., C.C.M., and A.C.D.; methodology, I.B., M.A., and A.C.D.; validation, C.S. and A.C.D.; investigation, I.B., M.A., C.C.M., A.B.S., and V.S.I.C.; writing—original draft preparation, I.B., M.A., and C.C.M.; writing—review and editing, C.S. and A.C.D.; visualization, I.B., M.A., and A.B.S.; supervision, C.S. and A.C.D. All authors have read and agreed to the published version of the manuscript.

**Funding:** This research received no external funding.

**Acknowledgments:** We would like to thank Vladimir Tudose for his help with the schematic representation in Figure 1.

**Conflicts of Interest:** The authors declare no conflict of interest.

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
