# Peer review of "Electrochemical Behavior of Rejected Dental Implants in Peri-Implantitis"

_coatings, doi:10.3390/coatings10030209_

Round 1
Reviewer 1 Report
This contribution is interesting and has original results. However, the paper must be improved in some aspects.
The introduction must be improved. There are articles that describe the effect of bacteria on corrosion and mechanical degradation of dental implants such as: Effect of the oral bacteria on the mechanical behavior of titanium dental implants”.The International Journal of Oral and Maxillofacial Implants. 27 (1) (2012), 64-68.
“Streptococcus sanguinis adhesion on titanium rough surfaces: effect of shot blasting particles”. A.Rodríguez-Hernández et al. Journal of Materials Science:Materials in Medicine 22 (8) (2011) 1913-1922. DOI 10.1007/s10856-011-4366-8.
The statistical study with standard deviation must be incorporate in the text with statistical significance deviations.
The different clinical cases must be discussed. The characterisation of the samples, details of the samples tested are not explained with detail.
The discussion of the results should be improved and explain the differences.
Author Response
1.This contribution is interesting and has original results. However, the paper must be improved in some aspects.
R1: Thank you for your comments. We tried to improve as much as we could.
2. The introduction must be improved. There are articles that describe the effect of bacteria on corrosion and mechanical degradation of dental implants such as: Effect of the oral bacteria on the mechanical behavior of titanium dental implants”.The International Journal of Oral and Maxillofacial Implants. 27 (1) (2012), 64-68.
“Streptococcus sanguinis adhesion on titanium rough surfaces: effect of shot blasting particles”. A.Rodríguez-Hernández et al. Journal of Materials Science:Materials in Medicine 22 (8) (2011) 1913-1922. DOI 10.1007/s10856-011-4366-8.
R2: Thank you for your suggestions. The introduction has been revised accordingly.
3. The statistical study with standard deviation must be incorporate in the text with statistical significance deviations.
R3: We performed statistical analysis on corrosion rates and polarization resistance measurements. The results are presented as summary statistics, i.e. means, standard deviations and medians. We also performed intergroup comparisons using non-parametric tests.
4. The different clinical cases must be discussed. The characterisation of the samples, details of the samples tested are not explained with detail.
R4: With this respect we detailed as much as possible.
5. The discussion of the results should be improved and explain the differences.
R5: The discussion was improved and the recorded differences were commented.
Reviewer 2 Report
The article is well written and can be considered for publication
however some minor concerns are needed:
some improvements in English language are needed the Authors should describe other references about perimplantitis, also in patients with systemic diseases, in particular the following papers: PubMed ID: 31781702 PubMed ID: 28696070 PubMed ID: 26238779 PubMed ID: 25955953 PubMed ID: 25106011
Author Response
- The article is well written and can be considered for publication. However some minor concerns are needed: some improvements in English language are needed the Authors should describe other references about perimplantitis, also in patients with systemic diseases, in particular the following papers: PubMed ID: 31781702 PubMed ID: 28696070 PubMed ID: 26238779 PubMed ID: 25955953 PubMed ID: 25106011
R1. Thank you for your comments. We included the suggested references and also corrected English language.
Reviewer 3 Report
The manuscript entitled "Electrochemical Behaviour of Rejected Dental Implants in Peri-Implantitis " deals with the assessment of the electrochemical stability of four dental implants based on titanium alloys using different methods.
The study is very interesting; however, some changes need to be addressed.
Abstract: Add more data concerning your findings.
Keywords: should be in alphabetical order, KEYWORDS should not contain the same words that are within the title of the text. Thus these should be changed appropriately
Introduction:
Line 41-42 Add a citation to this statement: “These properties include the implant microstructure and the composition and characteristics of its surfaces.” doi: 10.17219/acem/65069
Line 46-47 Add a citation to this statement “obtain a good stability of dental reconstructions all the factors that contribute to the oral 46 environment should be considered.” doi.org/10.1155/2019/2785302
Line 47-49 Add the newest citation to this statement “Salivary factors, microbial biofilms, and factors related to reconstructions are part of a unique, dynamic and complex system that influences short and long term prosthetic implant therapy [10]..” doi.org/10.3390/microorganisms7070189
M&M:
The sample collection is not well described. Describe the storage conditions of the samples prior to the experiment. Were all implants removed the day/two/… before the experiment? Where samples were kept/.
Do the implants have the same titanium grade?
Add the producers of the implants and the diameter/length.
Does the peri-implantitis was treated using lasers or scaling, these procedures can change the surface of implants and influencing the results of this study?
Line 88, 95 etc. Add a country of the producer “PGSTAT 302N (Metrohm)”
Line 99 etc. Add a country of the SEM, EDS unit producer. Add the Scanning electron microscope parameters e,g. acceleration, spot size.
Discussion
Add to this section the influence of various titanium grades on the results of this study.
Add to this section a paragraph describing the influence of bacteria on the process of periimplantitis.
References
Add correct abbreviations of the journals in the literature you cited.
Author Response
1. The manuscript entitled "Electrochemical Behaviour of Rejected Dental Implants in Peri-Implantitis " deals with the assessment of the electrochemical stability of four dental implants based on titanium alloys using different methods.
The study is very interesting; however, some changes need to be addressed.
R1. Thank you for your comments. We tried to perform all required changes.
2. Abstract: Add more data concerning your findings.
R2. The abstract has been revised accordingly.
3. Keywords: should be in alphabetical order, KEYWORDS should not contain the same words that are within the title of the text. Thus these should be changed appropriately
R3. The keywords were changed and ordered alphabetically.
4. Introduction:
Line 41-42 Add a citation to this statement: “These properties include the implant microstructure and the composition and characteristics of its surfaces.” doi: 10.17219/acem/65069
Line 46-47 Add a citation to this statement “obtain a good stability of dental reconstructions all the factors that contribute to the oral 46 environment should be considered.” doi.org/10.1155/2019/2785302
Line 47-49 Add the newest citation to this statement “Salivary factors, microbial biofilms, and factors related to reconstructions are part of a unique, dynamic and complex system that influences short and long term prosthetic implant therapy [10]..” doi.org/10.3390/microorganisms7070189
R4. These were done.
5. M&M:
The sample collection is not well described. Describe the storage conditions of the samples prior to the experiment. Were all implants removed the day/two/… before the experiment? Where samples were kept/.
R5.The sample collection, the storage conditions prior to the experiment were added and described in the manuscript.
6. Do the implants have the same titanium grade?
R6. As we do not know the implants producers, the titanium mass was determined through EDS analyses, as it can be seen in the results section. From the EDS results it can be noticed that all implants had different titanium grades. We commented this aspect in Discussion section.
7. Add the producers of the implants and the diameter/length.
R7.The manufacturers of the implants are not known, as the implants were inserted in other dental offices and there was no available data regarding the dental implants producers, from the previous patient’s medical history. The diameter and length were added in the manuscript in M&M section.
8. Does the peri-implantitis was treated using lasers or scaling, these procedures can change the surface of implants and influencing the results of this study?
R8. The peri-implantitis treatment was not conservative (local debridement and implant scaling) due to its severe status.The affected implants were explanted using rotary instruments (trephine drills) and specific implant removal kit.
9. Line 88, 95 etc. Add a country of the producer “PGSTAT 302N (Metrohm)” Line 99 etc. Add a country of the SEM, EDS unit producer. Add the Scanning electron microscope parameters e,g. acceleration, spot size.
R9. These were done.
10. Discussion. Add to this section the influence of various titanium grades on the results of this study. Add to this section a paragraph describing the influence of bacteria on the process of periimplantitis.
R10. Done.
11. References. Add correct abbreviations of the journals in the literature you cited.
R11. Hopefully now everything is fine. We used endnote program.
Round 2
Reviewer 1 Report
The new version of the contribution has been improved. However, some aspects should be consider:
- The trade mark of the implants studied should be described or the designs shown in a Figure. It is very important know if the implants are made with titanium cp or Ti-6Al-4V or other alloy.
- In the discussion is important to explain the influence of the abutment (usually of the other metal than the implant) which can produce an increase of the corrosion rate. The authors could describe -if they know- the nature of the abutment.
Author Response
1. Thank you for your comments. The requirements were addressed as follows:The trade mark of the implants studied should be described or the designs shown in a Figure. It is very important know if the implants are made with titanium cp or Ti-6Al-4V or other alloy.
R1: A schematic representation of the dental implants used in the study was added as Figure 1.The numbering of next figures has been modified accordingly. Information and comments regarding the implants materials nature have been added in the Results and Discussions sections. A correction has been also done in the Discussions section.
2. In the discussion is important to explain the influence of the abutment (usually of the other metal than the implant) which can produce an increase of the corrosion rate. The authors could describe -if they know- the nature of the abutment.
R2: Within the limits of our information, the design and material’s nature of the abutment have been discussed.
Reviewer 3 Report
Thank you for the review.
Author Response
Thank you, too